# Antimicrobial Prescribing Patterns in Patients with COVID-19 in Russian Multi-Field Hospitals in 2021: Results of the Global-PPS Project

**DOI:** 10.3390/tropicalmed7050075

**Published:** 2022-05-16

**Authors:** Sergey Avdeev, Svetlana Rachina, Yuliya Belkova, Roman Kozlov, Ann Versporten, Ines Pauwels, Herman Goossens, Elena Bochanova, Elena Elokhina, Ulyana Portnjagina, Olga Reshetko, Igor Sychev, Darya Strelkova

**Affiliations:** 1First Moscow State Medical University, 119435 Moscow, Russia; serg_avdeev@list.ru (S.A.); dashastrelkova@gmail.com (D.S.); 2Smolensk State Medical University, 214019 Smolensk, Russia; yuliya.belkova@antibiotic.ru (Y.B.); roman.kozlov@antibiotic.ru (R.K.); 3Vaccine and Infectious Disease Institute, University of Antwerp, 2610 Antwerp, Belgium; ann.versporten@uantwerpen.be (A.V.); ines.pauwels@uantwerpen.be (I.P.); herman.goossens@uza.be (H.G.); 4Krasnoyarsk State Medical University n.a. Professor V. F. Voyno-Yasenetsky, 660022 Krasnoyarsk, Russia; bochanova@list.ru; 5Omsk Regional Clinical Hospital, 644111 Omsk, Russia; elochina@yandex.ru; 6North-Eastern Federal University, 677007 Yakutsk, Russia; ulyana-nsk@mail.ru; 7Saratov State Medical University n.a. V. I. Razumovsky, 410000 Saratov, Russia; reshetko@yandex.ru; 8City Clinical Hospital n.a. S.S. Yudin, 117198 Moscow, Russia; sychevigor@mail.ru

**Keywords:** COVID-19, hospital, point prevalence survey, antimicrobials

## Abstract

The COVID-19 pandemic is a global public health challenge with understudied effects on antimicrobial usage. We aimed to analyze antimicrobial prescribing patterns in COVID-19 patients in Russian multi-field hospitals by means of the Global-PPS Project developed by the University of Antwerp. Out of 999 patients in COVID-19 wards in six hospitals surveyed in 2021, 51.3% received antimicrobials (79% in intensive care, 47.5% in medical wards). Systemic antivirals and antibiotics were prescribed to 31% and 35.1% of patients, respectively, and a combination of both to 14.1% of patients. The top antivirals administered were favipiravir (65%), remdesivir (19.2%), and umifenovir (15.8%); the top antibiotics were ceftriaxone (29.7%), levofloxacin (18%), and cefoperazone/sulbactam (10.4%). The vast majority of antibiotics was prescribed for treatment of pneumonia or COVID-19 infection (59.3% and 25.1%, respectively). Treatment was based on biomarker data in 42.7% of patients but was targeted only in 29.6% (6.7% for antibiotics). The rate of non-compliance with guidelines reached 16.6%. Antimicrobial prescribing patterns varied considerably in COVID-19 wards in Russian hospitals with groundlessly high rates of systemic antibiotics. Antimicrobial usage surveillance and stewardship should be applied to inpatient care during the COVID-19 pandemic.

## 1. Introduction

Since the first case of infection caused by severe acute respiratory syndrome coronavirus 2 (SARS-CoV-2), later known as COVID-19, was identified in Wuhan, China, in December 2019, the disease has affected more than 470 million people, with over 6 million fatalities worldwide as of March, 2022. In the Russian Federation, more than 17.6 million cases of COVID-19 with more than 365 thousand deaths have been registered since March 2020 when the first case of infection was reported [1].

The COVID-19 pandemic has had a severe impact on all aspects of the day-to-day lives of people globally, including devastating effects on healthcare systems. The escalation of antimicrobial resistance is another unavoidable consequence of this pandemic [2]. World Health Organization (WHO) guidelines as well as Russian national guidelines recommend that antibiotics should not be prescribed unless there is strong clinical suspicion of a bacterial infection [3,4]. At the same time, global and national studies provide evidence of wide and unjustified use of antibiotics in COVID-19 inpatients. Empirical broad-spectrum antibiotics have been reported to be administered to 60–100% of inpatients worldwide [5,6,7,8], despite the low incidence of secondary bacterial pulmonary infections [9,10]. A similar trend has been observed in Russia. According to a retrospective analysis of 1082 records of patients with laboratory-confirmed COVID-19 in 2020 in 17 regions of Russia (EGIDA-2020 study), up to 77% of hospitalized patients received antibiotics [11]. 

Still, information on antimicrobial consumption and prescribing in inpatients with COVID-19 in Russian hospitals remains sparse and understudied. Point prevalence studies (PPS) have established themselves as a convenient, low-cost, and at the same time standardized and validated tool for monitoring the prescribing of drugs in inpatients to aid antimicrobial stewardship (AMS) activities [12].

The present study aims to evaluate prescribing patterns of antimicrobials in COVID-19 patients in Russian multi-field hospitals and to quantify the prescribing in relation to quality indicators.

## 2. Materials and Methods

A point prevalence survey was conducted between June and December 2021, as a part of the Global-PPS Project [13], in six multi-field hospitals from different regions of the Russian Federation (Krasnoyarsk, Moscow, Omsk, Saratov, Smolensk and Yakutsk). We report antimicrobial drug (AMD) prescribing patterns in COVID-19 wards, including patients with suspected and proven COVID-19 infection receiving at least one antimicrobial for any indication. 

Each COVID-19 ward in the study was surveyed only once in a single day. All inpatients receiving an antimicrobial at 8 a.m. on the day of the PPS were included in the analysis. Data on the antimicrobial agents, reasons, indications, a set of quality indicators, and information on invasive device usage were collected. The prevalence of AMD prescription was calculated by dividing the number of patients treated with at least one antimicrobial agent over the total number of inpatients surveyed. 

Antimicrobials were classified according to the standardized WHO Anatomical Therapeutic Chemical (ATC) classification system [14] and the AWaRe system [15] and included: antibacterials for systemic use (ATC J01), antimycotics and antifungals for systemic use (J02 and D01BA), drugs for the treatment of tuberculosis (J04A), antibiotics used as intestinal anti-infectives (A07AA), antiprotozoals used as antibacterial agents, nitroimidazole derivatives (P01AB), antivirals for systemic use (J05), and antimalarials (P01B).

The prescription of antimicrobials in clinical practice was evaluated by means of quality indicators specified by the Global-PPS international study protocol:Compliance with local hospital guidelines,Documentation of indications for prescription of antimicrobial therapy,Documentation of stop/review dates,Targeted treatment based upon microbiological results,Treatment based upon the use of biomarker data (C-reactive protein, procalcitonin, or others).

All quality indicators, including compliance with hospital guidelines, were assessed by the local coordinator at each study site.

Full information on the method used is available on the website: www.global-pps.com (accessed on 1 April 2022).

The data were entered by the participating hospitals in the web-based application of the Global-PPS, with the database hosted at the University of Antwerp, Belgium. Data were analyzed by means of descriptive statistics.

## 3. Results

### 3.1. Characteristics of the Hospitals and Study Population

Of the 999 patients in 19 COVID-19 wards included in the survey, 88.1% were admitted to medical wards and 11.9% to intensive care units (ICU). The main characteristics of COVID-19 wards and the patient population with suspected or proven COVID-19 infection at each hospital are presented in Table 1.

Overall, 512 patients (51.3%) received at least one antimicrobial agent on the day of the PPS. The majority of those patents was female (60.9%), and the average age was 61.7 years. The rate of patients with confirmed previous hospitalization was as low as 13.5% (only 1.8% in the ICU), antibiotic treatment during the previous month was registered in 23.2% of patients, and surgery during current admission to hospital was performed in 6.1% of cases. The most common invasive device present on the day of the PPS was a peripheral vascular catheter (50.6% on average, variations 12.3–100%), followed by central vascular and indwelling urinary catheters (14.6% and 16.8%, respectively). Non-invasive ventilation was performed (11.3%) more often than invasive mechanical ventilation (4.1%). Although high-flow oxygen therapy was not registered as per the protocol, it had been commonly performed to non-ICU patients who had required respiratory support.

### 3.2. AMD Prescribing Patterns in COVID-19 Wards

A total of 512 patients (51.3%) received at least one antimicrobial on the day of PPS (Table 2). The average antiviral and antibiotic use prevalence reached 30.6% and 35.1%, respectively. Up to 13.2% of patients received drugs from both groups. Predictably, the AMD usage rate was higher in patients in ICUs (79% vs. 47.5%), predominantly due to the common prescribing of antibiotics (75.6% in ICU patients vs. 29.7% in patients with COVID-19 in medical wards). At the same time, antivirals were administered to ICU patients less commonly than to patients in medical wards (19.1% vs. 31.8% respectively).

Prominent diversity in antimicrobial use prevalence was noted at different hospitals, with the lowest rate at site #6 (34.3%) and the highest rate at site #4 (87.2%). Despite the significant variability of the results, it was possible to identify general trends in the prevalence of AMD prescribing, depending on the type of medical facilities. Thus, in medical wards, antiviral and antibiotic therapy was administered with similar frequencies (31.8% and 29.7%, respectively), whereas in ICUs, antiviral use prevalence was considerably lower (19.1%), probably due to the lack of parenteral antivirals (remdesivir) in some hospitals, while the rate of administration of antibiotics was twice as high (75.6%). 

The most common indications for therapeutic antibiotics (ATC J01, P01AB, A07AA) in patients in COVID-19 wards were pneumonia (59.5%) and COVID-19 infection (25.8%). It should be noted that although antibiotics in patients with suspected or proven COVID-19 infection were administered mostly for the treatment of secondary bacterial complications or bacterial co-infections, some hospitals reported “pneumonia” as an indication for antibacterial therapy, whereas others recorded these cases as “COVID-19”, without mentioning lower respiratory tract involvement. With that in mind, for the purpose of analysis in the current publication, we decided to combine administrations for both indications in a single entity “COVID-19/pneumonia” and use other parameters such as quality indicators to assess the appropriateness of therapy. 

The third most common indication for antibacterial therapy was *Clostridium difficile*-associated infection (CDI) (5.3%), while infections of other anatomical sites were responsible for 9.4% of antibiotic prescriptions (Table 3).

Only three systemic antivirals were prescribed to patients with suspected or proven COVID-19 infection in Russian hospitals in 2021. Overall, favipiravir (64.6%) was the most often prescribed antiviral drug, followed by remdesivir (20%) and umifenovir (15.4%). It should be noted that remdesivir was available only in two out of six hospitals (site #5 and #6). 

The top systemic antibiotics, administered to patients with lower respiratory tract infections in COVID-19 wards in Russian hospitals, included ceftriaxone (31.5% on average), levofloxacin (19.1%), and cefoperazone/sulbactam (11%). Less commonly, inhibitor protected amoxicillin, cefepime, and cefepime/sulbactam were used (5.6%, 5.4%, and 5.1%, respectively). Substantial diversity in the prevalence of prescribing of different antibiotics was noted among the hospitals. In hospital #5, the top three administered antibiotics were cefepime/sulbactam (22.1%), cefepime (17.4%), and levofloxacin and ampicillin/sulbactam (14% each), while in hospital #2, half of the administered antibiotics represented inhibitor-protected amoxicillin (Table 4). 

It is worth mentioning that the total share of third- to fifth-generation cephalosporins was as high as 53.3% (variations from 23.5% at site #2 to 71.9% at site #1), fluoroquinolones —22.4% (variations from 0% at site #1 and #2 to 40.9% at site #6), and carbapenems—8.3% (variations from 3.3% at site #3 to 11.8% at site #2).

While the type of antivirals prescribed for the treatment of COVID-19 infection in medical wards and in ICUs was almost similar, the type of prescribed antibiotics varied substantially (Figure 1). Ceftriaxone and aminopenicillins were administered more frequently in medical wards, while cephalosporins with antipseudomonal activity, carbapenems, and levofloxacin were predictably more common in ICUs.

### 3.3. Key Patterns and Quality Indicators of Systemic AMD Prescribing for “COVID-19/Pneumonia”

The analysis of key patterns and quality indicators of systemic AMD prescribing was limited to the major group of patients with suspected or proven “COVID-19/pneumonia”. In this subgroup, antimicrobial treatment was based on biomarkers in 42.7% of cases (58.6% in ICUs vs. 39.2% in medical wards). The most common biomarker used to support therapeutic decision was C-reactive protein (25.5% on average); procalcitonin was used only in 6.6% of patients in medical wards and 12.6% in ICUs. Culture tests were performed only in 14.1% of patients (Table 5).

Antibiotic prescribing for suspected or proven bacterial pneumonia was predominantly empirical in patients with COVID-19 infection regardless of the type of ward (70.4% in total, 71% in ICUs, 70.2% in medical wards). Targeted therapy was more common for antivirals (58.2% in total), especially in ICUs (86.7%) but unreasonably low for antibiotics (6.7% in total, 4.2% in medical wards). Antivirals complied with the hospital guidelines in 100% of prescriptions with indications for treatment (“COVID-19”), and stop/review dates were recorded in 96.6% of cases. At the same time, for antibiotics, the non-compliance rate was as high as 29.9%, with prominent diversity among the hospitals (from 4.5% at site #6 to 69.8% at site #5). Indications for treatment and stop/review dates were recorded in 79% and 76%, respectively.

The majority of prescribed antibiotics belonged to the “watch” group of the AWaRe classification (73.3%), although at site #2 the “access” group prevailed (58.8%). “Reserve” antibiotics were prescribed rarely (4.3% on average, variations from 0% at sites #1, #3, and #6 to 9.4% at site #4); the share of “not recommended” drugs, such as cefoperazone/sulbactam, was relatively high at sites #1 and #4 (23.1% and 17.9%, respectively). 

## 4. Discussion

In this article, we present the results of a multi-center study of antimicrobial treatment practice in patients with suspected or proven COVID-19 in designated clinical areas, including critical care, in six Russian hospitals. The study was carried out in the second half of 2021 as a part of the international Global-PPS Project [13] using the conventional point prevalence study methodology [16]. PPS is a universal tool that allows information to be obtained on the use of AMD in hospitalized patients to reveal the main problems and develop targeted measures as a part of local AMS programs and to monitor the effectiveness of their implementation [12,17]. The results of the study will provide a basis for the development of appropriate stewardship activities, tailored according to local practices for each multi-field hospital in the project in the COVID-19 pandemic period and thereafter.

It should be noted that COVID-19 infection not only has led to myriad fatalities directly [1], but it has also contributed to the development of another worldwide crisis, namely the crisis of antibiotic resistance. The long-term influence of COVID-19 on resistance has become a concern due to the widespread use of antibiotics and disinfectants in patients infected with SARS-CoV-2 [18,19,20]. Emergent antibiotic resistance following the global response to COVID-19 has already been described as “the silent pandemic” [2], which emphasizes the importance of prudent use of AMD in this group of patients. The preliminary data on COVID-19 influence on epidemiology of nosocomial infections and antimicrobial resistance in Russian hospitals were diverse and inconclusive [21], but a study performed in a hematological center in Kazakhstan showed a prominent increase in the prevalence of resistant nosocomial isolates; the share of *Pseudomonas aeruginosa* carbapenemase-producing strains increased from 0% in the pre-pandemic period to 30.4%; the share of methicillin-resistant *Staphylococcus aureus* increased from 11.8% to 35%, respectively [22]. The selection of resistance to antibiotics combined with the lack of perspectives for the introduction of new drugs in clinical practice in the near future highlights the urgent need to reassess the practice of AMD prescribing. 

Another potential problem following overuse of antimicrobials in COVID-19 patients is the growing burden of CDI. Indeed, the elevated prevalence of CDI was observed in a USA hospital from January–February until March–April 2020 [23]. An increase in the CDI rates from 2.6% to 10.9% during the COVID-19 pandemic was reported in a Polish hospital [24], and from 0.7 to 12.3 per 10,000 patient days in eight Italian acute-care hospitals admitting COVID-19 patients [25]. In Russia, during the period between 2019 and 2020, an increase in the rate of circulating toxin-producing *C. difficile* strains was revealed in patients with colitis and enterocolitis (from 4% to 11% in adults and from 5% to 16% in children), which could also be the result of antibiotics overuse [26].

Approaches to antimicrobial therapy in COVID-19 patients have been changing throughout the whole period of the pandemic due to the lack of effective regimens and uncertainty about the rate of bacterial complications. In Russia, as of March 2022, 15 revisions of temporary national guidelines on prophylaxis, diagnostics, and management of COVID-19 have been introduced. The list of antiviral options in the latest revisions included favipiravir, molnupiravir, remdesivir, and umifenovir as well as neutralizing immunoglobulins against SARS-CoV-2 (casirivimab, imdevimab, bamlanivimab, etesevimab, sotrovimab, etc.). Antibacterial agents are to be given only to patients with solid evidence of bacterial infection, and the list of options is consistent with the recommendations for the treatment of bacterial pneumonia in a non-COVID-19 population [4]. This approach is aligned with WHO guidelines recommending that antibiotics not be prescribed unless there is clinical suspicion of a bacterial infection [3].

The results of our study revealed that overall, 51.3% of patients received at least one antimicrobial agent on the day of PPS. On average, about 1/3 of patients received either antivirals or antibiotics, and 14.1% received a combination of both. 

The rate of patients who received antiviral therapy on the day of the PPS (31%) was vastly different from the 92% observed in the retrospective EGIDA-2020 study of 1082 records of patients with laboratory-confirmed COVID-19 in 17 regions of Russia [11]. This gap can be partially explained by different methodologies of the projects. The PPS did not provide data for the whole period of hospitalization but allowed us to assess the proportion of patients with medical treatment in a hospital in a single day. Another matter is the diversity in approaches to COVID-19 management in 2020 and in 2021. The lack of effective options for the treatment in a pandemic with high mortality rates promoted wide use of any potentially effective agents, which was reflected in the guidelines during that time period [11].

In the EGIDA-2020 study, only patients with laboratory confirmed SARS-CoV-2 infection were included, and all of them had treatment options that later failed to prove efficacy for the treatment of COVID-19 (hydroxychloroquine, imidazolylethanamide pentandioic acid, lopinavir + ritonavir combination, umifenovir, and interferons) [11]. In another Russian retrospective study performed in September–December 2020 in the population of hospitalized patients with COVID-19 (46% laboratory confirmed), 40% of patients had received antiviral therapy before admission (umifenovir, interferons, imidazolylethanamide pentandioic acid, favipiravir, etc.), while only 4% of patients received antiviral (remdesivir) during hospitalization [27]. 

In our study, only three antiviral options were prescribed (favipiravir—65%, remdesivir—19.2%, and umifenovir—15.8% on average). It is of importance that remdesivir was available only in two hospitals, with the rate of prescription amounting to up to 60%, whereas SARS-CoV-2 neutralizing immunoglobulins were not available in any hospital at that point in time. Diversity in the prescribing of antivirals in different hospitals in the project probably resulted from uncertainty in the effectiveness of treatment options as well as limited availability of the drugs.

Antibacterial agents were administered to 35.1% of patients on average in the study sites on the day of PPS. Prescribing in ICUs was heterogeneous but overall consistently high across healthcare settings (75.6% on average, variations from 52.7% to 100%), while in medical wards, antibiotics were given less commonly (29.7% on average, variations from 19% to 62%). These rates are lower in comparison with the global data (60–100% [5,6,7,8]) and the data from the Russian EGIDA-2020 study [11] that revealed 77% antibiotic coverage in COVID-19 inpatients in 2020, which can be explained by the methodological differences of the studies as well as the global trend to restrict antibacterial prescribing in COVID-19 patients. By now, it is a well-known fact that bacterial infections are relatively rare in patients with COVID-19 outside of intensive care [9,10], which makes antibacterial therapy unjustified in certain portion of patients and provides an opportunity for AMS interventions.

The methodology of PPS did not allow us to directly assess the rate of bacterial complications in inpatients with COVID-19, but the prevalence of nosocomial infection risk factors in our study was relatively low; only 13.5% of patients treated with at least one antimicrobial had previous hospitalization, 23.2% had antibiotic treatment in the previous 3-month period, and only 6.1% underwent surgery during the current admission to the hospital. The total share of patients with invasive and non-invasive ventilation did not exceed 15%, although the number of patients with high-flow oxygen therapy not registered per the protocol was much higher. In addition, low rates of targeted antibacterial therapy (6.7%) and culture tests performed (14.1%) were revealed. These data combined raise an issue regarding the rationale for antibiotic therapy in patients with COVID-19 in the participating hospitals and provide another target for AMS activities.

Treatment was based on biomarker data in 42.7% of patients, including C-reactive protein in 25.5% and procalcitonin tests in 7.7% of cases. Suggested as a biomarker to rule out bacterial complications/co-infections in patients with COVID-19 at the onset of the pandemic, C-reactive protein later proved to have low diagnostic value, being non-specific for bacterial infection and elevated in many hospitalized patients with COVID-19 [28,29]. The more prognostically valuable procalcitonin test is collected on demand in Russian clinical practice, mostly in patients with severe COVID-19 admitted to the ICU to exclude bacterial co-infection and/or superinfection with a level ≥0.5 mcg/L as a cutoff [4].

The list of antibacterial agents administered corresponded to the national guidelines for the treatment of bacterial pneumonia (either community-acquired or nosocomial). The high frequency of cephalosporin prescribing in Russian hospitals is a well-known fact [30]. Not surprisingly, ceftriaxone was the leading option, followed by levofloxacin (29.7% and 18% of prescriptions on average, respectively). According to the WHO AWaRe classification of antibiotics [15], third-generation cephalosporins belong to the “watch” group and are considered as key targets of stewardship programs and monitoring; fluoroquinolones fall into the same category. Overall, 73.3% of antibiotics administered to patients with COVID-19 in our study centers belonged to the “watch” group, but the combined rate of “reserve” and “not recommended” drugs was also noticeable (15.6% on average, up to 27.3% at some sites). The prominent diversity of antibiotic prescribing in different hospitals is possibly related to the local epidemiology of infections and the list of available treatment options. We also observed high rates of antibacterial therapy non-compliance to hospital guidelines at some sites, including overuse of broad-spectrum antibiotics, which contradicts Russian national guidelines strictly recommending antibiotics only to patients with strong clinical suspicion of a bacterial infection [4]. Although the PPS methodology does not allow us to conclude with certainty on the rationale behind prescribing practice at the study sites, we can speculate that overuse of broad-spectrum antibiotics may be related in part to common use of immunosuppressive therapy, including high doses and prolonged courses of glucocorticoids, in hospitalized adults with COVID-19. This may have contributed to higher rates of bacterial superinfections caused by nosocomial multidrug resistant pathogens, which are common in Russian hospitals [31,32,33]. Non-compliance to guidelines as well as the low rate of recording of the antimicrobials’ stop/review dates should also be covered in the hospitals’ local AMS programs.

It is necessary to mention the limitations inherent in the study and directly stemming from its methodology. The data obtained during the project were generalized and provided no incites at the individual patient level. The study did not allow us to account for all antimicrobials administered to patients during the period of hospitalization nor to assess the number of inpatients who had antibacterial therapy as a whole. Another limitation was diagnosis-coding inconsistency, which did not allow us to differentiate between COVID-19 infection and bacterial co-infections in patients who had antibacterial therapy.

Although the Global-PPS Project methodology did not allow us to evaluate the rationality of drug prescribing in individual patients, it revealed the practice of antibiotic overuse in patients with COVID-19 as well as issues related to the choice of agents, including wide use of third- and fourth-generation cephalosporins and fluoroquinolones. This needs to be reassessed because of the negative influence on selection of resistance and CDI epidemiology. Due to the simplicity and standardization of data collection and the use of a web-based data entry environment for the consolidation, validation, and reporting of data, PPS can be used to increase the awareness of specialists about the situation in health facilities during the COVID-19 pandemic and thereafter. The Global-PPS as such provides considerable aid in the development of tailored strategies for antimicrobial stewardship, which can be re-assessed through repeated PPS.

## 5. Conclusions

Antimicrobial prescribing patterns in patients with COVID-19 vary considerably among hospitals in Russia. The prescribing of systemic antibiotics is high, especially in ICUs, with a liberal share of broad-spectrum agents such as third- to fourth-generation cephalosporins, fluoroquinolones, and carbapenems. The high rate of antibacterial therapy non-compliance to guidelines is still an issue at some hospitals. Antimicrobial usage surveillance and stewardship should be applied to inpatient care during the COVID-19 pandemic and thereafter.

## Figures and Tables

**Figure 1 tropicalmed-07-00075-f001:**
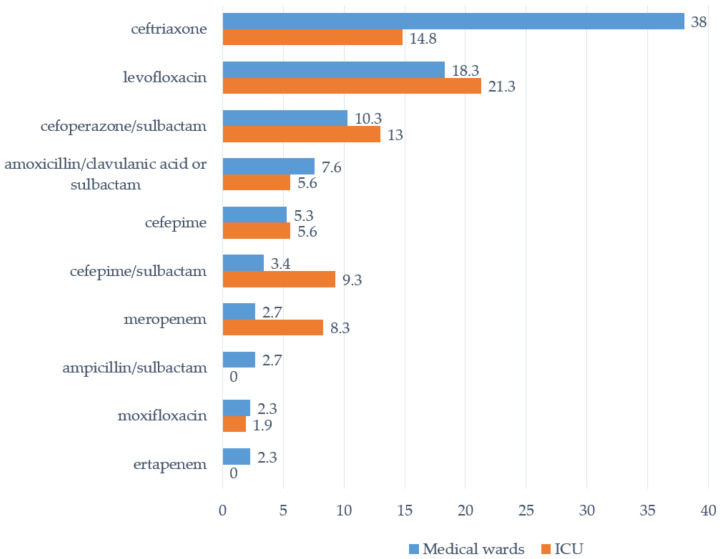
Top 10 systemic antibiotics administered to patients with “COVID-19/pneumonia” in COVID-19 medical wards and ICUs of 6 Russian hospitals, % of prescriptions.

**Table 1 tropicalmed-07-00075-t001:** Characteristics of the hospitals and detailed information on patients admitted to COVID-19 wards and treated with at least one antimicrobial.

Characteristics	Site #1	Site #2	Site #3	Site #4	Site #5	Site #6	Total/Average
COVID-19 wards surveyed, n	1	3	2	4	8	1	19
medical wards	0	1	0	2	5	1	9
ICU	0	0	0	2	3	0	5
mixed wards (medical and intensive care beds)	1	2	2	0	0	0	5
Patients at the COVID-19 wards on the day of PPS, n	110	67	313	133	306	70	999
Patients receiving antimicrobials on the day of PPS, n	70	35	106	116	161	24	512
Sex, % of male	35.7	68.6	34	39.7	36	45.8	39.1
Average age, years	61.6 ± 16.1	49 ± 16.5	53.2 ± 16.8	69 ± 15.6	70.8 ± 13.6	60.5 ± 12.9	61.7 ± 17
Previous hospitalization, %							
yes, ICU	0	2.9	5.7	0	1.2	0	1.8
yes, other	0	31.4	23.6	0	13.7	8.3	11.7
no	0	65.7	4.7	81	83.2	83.3	53.9
unknown	100	0	66	19	1.9	8.3	32.6
Previous antibiotic treatment, %							
yes	14.3	20	39.6	6	30.4	16.7	23.2
no	10	57.1	58.5	89.7	60.9	79.2	60.5
unknown	75.7	22.9	1.9	4.3	8.7	4.2	16.2
Surgery during current admission in hospital, %							
yes	0	51.4	8.5	0.9	1.2	4.2	6.1
no	100	48.6	91.5	97.4	98.8	91.7	93.4
unknown	0	0	0	1.7	0	4.2	0.6
Invasive device present on the day of PPS *, %							
indwelling urinary catheter	20	28.6	11.3	14.7	18	16.7	16.8
peripheral vascular catheter	54.3	31.4	12.3	65.5	60.2	100	50.6
central vascular catheter	0	25.7	19.8	19.8	13	4.2	14.6
non-invasive ventilation (CPAP, BiPAP, etc.) **	11.4	2.9	14.2	7.8	14.9	4.2	11.3
invasive mechanical ventilation	4.3	2.9	4.7	4.3	4.3	0	4.1
inserted tubes and drains	0	11.4	2.8	0.9	0	0	1.6

* Calculated as number of admitted patients on antimicrobials with an invasive device/all admitted patients on antimicrobials. ** CPAP—constant positive airway pressure, BiPAP—bi-level positive airway pressure.

**Table 2 tropicalmed-07-00075-t002:** AMD use prevalence * in patients in COVID-19 wards on the day of PPS.

Characteristics	Site #1	Site #2	Site #3	Site #4	Site #5	Site #6	Total/Average
Patients in the COVID-19 wards on the day of PPS, n	110	67	313	133	306	70	999
medical beds	100	67	284	108	251	70	880
intensive care beds	10	0	29	25	55	-	119
Antimicrobial prevalence, %	63.6	52.2	33.9	87.2	52.6	34.3	51.3
medical beds	60	52.2	28.2	84.3	51	34.3	47.5
intensive care beds	100	-	89.7	100	60	-	79
Antiviral prevalence, %	42.7	44.8	17.9	56.4	31.7	7.1	31
medical beds	44	44.8	17.3	58.3	35.5	7.1	31.8
intensive care beds	30	-	24.1	48	14.5	-	25.2
Antibiotic prevalence, %	36.4	32.8	25.6	69.2	31.4	30	35.1
medical beds	30	32.8	19	62	26.7	30	29.7
intensive care beds	100	-	89.7	100	52.7	-	75.6
Combination of antivirals and antibiotics, %	15.5	25.4	9.9	38.3	7.8	1.4	14.1
medical beds	14	25.4	8.1	36.1	7.6	1.4	12.8
intensive care beds	30	-	27.6	48	9.1	-	23.5

* Prevalence is calculated as the number of patients on at least one antimicrobial or antiviral or antibiotic/number admitted patients on the day of the PPS.

**Table 3 tropicalmed-07-00075-t003:** The most common indications for therapeutic antibiotics (ATC J01, P01AB, A07AA) in patients in COVID-19 wards, %.

Indication	Site #1	Site #2	Site #3	Site #4	Site #5	Site #6	Average
Pneumonia or lower respiratory tract infection	92.9	69.6	85.7	41.5	33	100	59.3
COVID-19 infection	0	0	0	41.5	47.6	0	25.1
*C. difficile*-associated infection	0	4.3	3.8	5.7	8.7	0	5
Upper urinary tract infection	2.4	0	2.9	0	8.7	0	2.8
Skin and soft tissue infection	0	13	1.9	0	0	0	1.1
Sepsis/bacteremia with no clear anatomic site	4.8	0	1	8.8	0	0	3.7
Bronchitis	0	0	0	1.9	0	0	0.7
Intra-abdominal infection	0	0	1.9	0	0	0	0.4
Obstetric/gynecological infection	0	0	1.9	0	0	0	0.4
Lower urinary tract infection	0	0	1	0	1	0	0.4
Other	0	13	0	0.6	1	0	1.1

**Table 4 tropicalmed-07-00075-t004:** Overall proportional use of systemic AMD (antibiotics and antivirals) for the treatment of patients with “COVID-19/pneumonia”, % of prescriptions.

Antibacterials	Site #1	Site #2	Site #3	Site #4	Site #5	Site #6	Average
Antivirals, %							
favipiravir	100	40	78.6	69.3	42.6	40	65
remdesivir	0	0	0	0	57.4	60	19.2
umifenovir	0	60	21.4	30.7	0	0	15.8
Antibiotics, %							
ceftriaxone	38.5	17.6	54.4	35.6	0,0	36.4	31.5
levofloxacin	0.0	0.0	25.6	22.9	14.0	40.9	19.1
cefoperazone/sulbactam	23.1	0.0	5.6	17.8	2.3	18.2	11.0
amoxicillin/clavulanic acid + amoxicillin/sulbactam	17.9	52.9	3.3	0.8	1.2	0.0	5.6
cefepime	10.3	5.9	0.0	0.0	17.4	0.0	5.4
cefepime/sulbactam	0.0	0.0	0.0	0.0	22.1	0.0	5.1
meropenem	7.7	11.8	3.3	1.7	5.8	4.5	4.3
ampicillin/sulbactam	0.0	0.0	0.0	0.0	14.0	0.0	3.2
imipenem	0.0	0.0	0.0	7.6	0.0	0.0	2.4
moxifloxacin	0	0	0	1.7	7	0	2.2
ertapenem	0.0	0.0	0.0	0.8	5.8	0.0	1.6
amikacin	0.0	0.0	2.2	0.0	3.5	0.0	1.3
linezolid	0.0	5.9	0.0	2.5	1.2	0.0	1.3
other	2.6	5.9	5.6	8.5	5.8	0.0	5.9

**Table 5 tropicalmed-07-00075-t005:** Key patterns and quality indicators of systemic AMD prescribing for the treatment of “COVID-19/pneumonia”.

Patterns	Site #1	Site #2	Site #3	Site #4	Site #5	Site #6	ICU Wards	Medical Wards	Average
Treatment based on biomarker data, % of patients	52.2	80.8	40.4	56.5	17.4	75	58.6	39.2	42.7
C-reactive protein *	34.8	65.4	35.4	38.3	2	0	32.2	24.1	25.5
white blood cells	17.4	3.8	0	8.7	8.7	33.3	13.8	8.1	9.1
procalcitonin	0	3.8	5.1	9.6	6.7	41.7	12.6	6.6	7.7
Culture test performed, % of patients	0	92.3	5.1	2.6	24.2	0	11.5	14.7	14.1
Quality indicators, % of prescriptions									
Targeted therapy	10.5	32.4	44.5	7.8	53.9	0	29	29.8	29.6
antivirals	19.1	40	96.4	17.3	94.7	0	86.7	55.1	58.2
antibiotics	0	23.5	12.2	1.7	9.3	0	13	4.2	6.7
Compliance with the hospital guidelines	81.4	97.3	100	82.8	66.7	96.3	79.0	84.5	83.4
antivirals	100	100	100	100	100	100	100	100	100
antibiotics	59	94.1	100	71.8	30.2	95.5	73.1	68.8	70.1
Indication for treatment was recorded	96.5	97.3	100	81.3	79.4	96.3	85.5	89.1	88.3
antivirals	100	100	100	100	100	100	100	100	100
antibiotics	92.3	94.1	100	69.2	57	95.5	81.5	77.9	79.0
Stop/review date documented	66.3	100	93.2	90.1	92.2	0	87.7	84.5	85.2
antivirals	97.9	100	100	100	95.7	0	100	96.3	96.6
antibiotics	28.2	100	88.9	83.8	88.4	0	84.3	72.6	76
Prescribed antibiotics according to AWaRe classification, % of prescriptions									
access	20.5	58.8	6.7	0.9	18.6	0	10.2	11.4	11.1
watch	56.4	35.3	86.7	71.8	74.4	81.8	70.4	74.5	73.3
reserve	0	5.9	0	9.4	4.7	0	5.6	3.8	4.3
not recommended	23.1	0	6.7	17.9	2.3	18.2	13.9	10.3	11.3

*—Median value of C-reactive protein at all study sites was 58.2 mg/L.

## Data Availability

The data presented in this study are available on request from the corresponding author. The data are not publicly available due to policy rules which are made by the Global-PPS team in order to safeguard confidentiality principles (privacy and ethical) and ensure the sustainability of the network.

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
