# Peer review of "Antimicrobial Prescribing Patterns in Patients with COVID-19 in Russian Multi-Field Hospitals in 2021: Results of the Global-PPS Project"

_tropicalmed, 2022, doi:10.3390/tropicalmed7050075_

Round 1

Reviewer 1 Report

Overall, very well written article in a rather current topic.

Introduction, methods and results are well described, and discussion offers very good interpretation. Additional consideration of possible explanations for the antibiotics prescribing variations could be beneficial (such as antibiotic shortages, price, availability, etc.), but not critical to the already great discussion.

The paper could further benefit from minor language revisions.

Reviewer 2 Report

The authors of the manuscript "Antimicrobial Prescribing Patterns in Patients with COVID-19 in Russian Multi-field Hospitals in 2021: Results of the Global-
PPS Project" did a very good contribution for the understanding of COVID-19 related increase of antimicrobial resistance. They successfully draw attention by PPS method on the lingering problem of unjustified, uncontrolled use of antimicrobials. As indicated, it leads to worsening the antimicrobial resistance phenomen. They call for stewardship and surveillance of antimicrobial use globally.

The PPS method helped to identify the problem of AMR during COVID-19. 

However, missing information of the study opens up new questions.

The authors state rightfully the missing patient information and even clearcut diagnosis in many cases.

The information is surely a result of a health system, trying to save many lives as possible under pressure during the COVID-19 pandemic.

It would have been interesting to have another PPS in the same settings after a given time of around 2-3 weeks. This would have allowed to estimate the outcomes of the treatments by antimicrobials in more enhanced way.

However, the authors state, that, this is the methodology of the study to pinpoint.

At line 251, pseudomembranous colitis is not a result of antibiotic-associated adverse drug reaction. It happens by selection of resistant strains though the antibiotic. This is surely known by the authors but the sentence should be corrected in order to avoid misunderstanding for the general reader.

The manuscript is well-written, interesting and fit for publication after minor revision.

Reviewer 3 Report

The manuscript by Avdeev et al describes a point prevalence study among 6 russian hospitals about antimicrobial and antiviral intake in hospitalizes patients with COVID-19.

Another study more to conclude that antimicrobial consumptionnis high although bacterial coinfection is low.

The study lacks novelty but is well-conducted and the manuscript well-written.

The authors should explain more about biomarkers and biomarker-appropriate treatment; it seems that they just detected the proportion of patients in which crp and pct was measured but what if this measurement was low. Perhaps it would be better to give antibiotic prevalence with specific cutoffs of the biomarker.

Second, the authors report high adherence to hospital- and national guidelines. National guidelines recommend antimicrobials so offen? I am not sure about that!Last, are there any data about MDR detection rate in these patients?

Round 2

Reviewer 3 Report

I regret to say the authors haven’t addressed my comments appropriately.